# Examining the Complex (Curvilinear and Contingent) Associations between Social Distancing Compliance and Subjective Health during a Global Health Crisis

**DOI:** 10.3390/ijerph192316058

**Published:** 2022-11-30

**Authors:** Jong Hyun Jung, Kyung Won Choi, Harris Hyun-soo Kim

**Affiliations:** 1Department of Sociology, Sungkyunkwan University, Seoul 03063, Republic of Korea; 2Department of Sociology, University of Chicago, Chicago, IL 60637, USA; 3Department of Sociology, Ewha Womans University, 52 Ewhayeodae-gil, Seodaemun-gu, Seoul 03760, Republic of Korea

**Keywords:** COVID-19, social distancing, subjective health, steeling hypothesis

## Abstract

Objectives: This study investigated a potential curvilinear link between social distancing behavior and subjective health in later life. It also evaluated whether food insecurity and community social capital moderated the focal relationship. Methods: Data were drawn from three waves of the COVID Impact Survey (N = 19,234). Mixed-effects models were fitted. Results: Social distancing has a non-monotonic (U-shaped) relationship with subjective health, i.e., individuals with low and high levels of social distancing show relatively better health. Moreover, the negative linear relationship between social distancing and health is weaker among people suffering from food insecurity as well as those living in communities with lower stocks of social capital. Discussion: This study sheds new light on the health implications of social distancing during the pandemic. Our findings dovetail with the steeling hypothesis, i.e., that social distancing is less harmful for U.S. older adults exposed to prior stressful or vulnerable conditions.

## 1. Introduction

A voluminous body of research in gerontology has examined the implications of social distancing during the COVID-19 pandemic for older adults’ health and well-being [1,2,3]. Although the bulk of evidence points to deleterious effects of social distancing [4,5,6,7], some findings show a null association [2,3] while others indicate an even a positive relationship with health and well-being in later life [1]. These divergent results raise the possibility of a non-monotonic link between social distancing and health. That is, there may be a curvilinear relationship. Arguments for a potential curvilinear relationship suggest that it is a risk factor but only to a certain extent. Studies have highlighted that COVID-19-related protective behavior can take a major toll on mental health as it results in social disconnection and loneliness [8]. Yet, there is a relatively limited amount of research concerning physical health. Given that physical and mental health conditions are intricately related [9], higher levels of social distancing can signal worse or poor self-rated health.

To the contrary, however, they may also indicate better assessment of one’s own health status because of its protection against the potentially lethal viral transmission. While the initial period of physically, hence socially, isolating oneself can be detrimental, over time individuals under self-imposed quarantine may reap the benefits of it. That is, people who are ‘highly’ engaged in social distancing would be less vulnerable to the COVID-19 infection. Thus, by comparing themselves to infected individuals, they may gain a sense of assurance which can boost subjective health perception [10]. In this sense, we might observe better health among people who more faithfully adhere to anti-pandemic guidelines/mandates. Taken together, then, these ideas suggest a potentially curvilinear link between social distancing and subjective health resembling a U-shape curve. In the scholarship, however, it is invariably assumed that this focal link is linear: the greater the level of social distancing, the worse the health outcome. According to systematic reviews [11,12], in fact, not a single finding was discussed in relation to non-monotonicity. The present study’s first objective, therefore, is to empirically examine this possibility.

Our second objective is to probe whether and how the harmful impact of social distancing may vary across two moderators, measures of vulnerability and resource. As Bierman and his colleagues recommend [13] “subsequent research into the health effects of the COVID-19 response should take care to consider how antecedent stressful conditions may have had further health effects due to increased vulnerabilities (and worse health perceptions) brought on by social distancing measures.” We respond to this call by focusing on where experienced food insecurity at the individual level (i.e., personal vulnerability) and community social capital at the county level (i.e., collective resource) condition the main association between social distancing and subjective health in later life. On the one hand, as prior research has shown, food insecurity is a potent stressor with deleterious consequences particularly for the old, or more ‘vulnerable,’ population [14,15,16,17]. On the other, empirical evidence points to health advantages of ecological or contextual social capital, i.e., community-level resources such as norms of reciprocity, collective identity, and mutual support [18,19,20,21,22,23]. We build on this line of research by conceptualizing community social capital as a collective resource in buffering the negative effect of social distancing.

### 1.1. Sensitizing Mechanism

We articulate and test two competing perspectives concerning the mechanisms underlying the moderating roles of food insecurity and community social capital. Consistent with prior research, we anticipate a generally negative association between social distancing and self-rated health [6,24]. However, we contend that this focal association is contingent across measures of food insecurity and community social capital. One potential explanation or mechanism underscores a process referred to as sensitizing, whereby initial exposure to stressors may sensitize individuals to subsequent stressors [25]. Being exposed to adverse factors is said to deplete psychosocial resources that could otherwise help reduce the harmful impact of other stressors [26]. Hence, individuals under stressful conditions may experience a heightened sensitivity to additional stressors, which in turn exacerbates their already compromised health [27]. Cast within the context of our study, this sensitizing view suggests that health-damaging effects of social distancing become amplified for those with higher exposure to food insecurity and *lower* access to community social capital.

One of the most consistent findings from national surveys is that the COVID-19 pandemic has worsened food shortages among low-income American households leading to dire situations [15,17,28,29]. Victims of food insecurity struggle financially. Additionally, having inadequate financial resources can severely undermine adaptive processes such as problem solving that are necessary to mitigate the effects of social distancing. Moreover, people in areas with low levels of social capital may lack communal resources including social cohesiveness, interpersonal trust, and mutual dependence [18,20,30,31,32]. In such circumstances, it becomes a huge challenge to solicit social support in times of need, especially during a pandemic, thereby intensifying the feelings of disconnectedness and loneliness caused by social distancing. Thus, it is likely that a residential context with lower stocks of social capital would exacerbate the negative impact of social distancing.

### 1.2. Steeling Mechanism

By contrast, the second underlying mechanism centers on the idea of steeling, where the initial stressors are thought to strengthen, not undermine, resilience against latter ones including external shocks such as a pandemic [33,34]. When individuals are confronted with stressful conditions, it may stimulate the development of personal resources to manage these stressors. This successful coping may foster one’s ability to adapt to and cope with subsequent ones [35]. Hence, exposure to stressors may produce a solidifying, or toughening, effect whereby people become relatively more, not less, resistant to future adverse experiences [36,37,38]. Applied to the present study, the so-called steeling view implies that noxious health effects of social distancing can be attenuated for persons who have experienced greater food insecurity and have access to limited social capital. That is, individuals suffering from food shortage may possess coping strategies that enable them to better manage the stressful demands of social distancing measures. Conversely, economically more advantaged (i.e., psychologically less resilient) individuals without experienced food insecurity may have a tougher time dealing with them.

In addition, the residents of communities with lower levels of social capital may have had similar occasions or reasons to acquire personal resilience against external shocks (e.g., natural disasters). In high-social capital context, by definition, members are cohesively interconnected, actively involved with one another, and thus more willing to assist as well as receive support from others during times of crises [39,40]. As a result, on average, they may not be fully equipped with the necessary coping skills to independently combat social isolation caused by the pandemic. In other words, residency in a more resourceful community can, to some extent, undermine individual adaptability to COVID-19 guidelines and mandates [41]. This would not be the case, however, for residents of places characterized by an insufficient stock of social capital—where they would be forced to mostly fend for themselves in the absence of material and emotional support by others. An unintended consequence, therefore, is that they may possess better coping skills at their disposal to confront the deleterious challenges of anti-coronavirus social distancing and other preventive measures.

### 1.3. Research Hypotheses

In sum, the findings and perspectives outlined above provide a basis for following hypotheses to be tested: 

**Hypothesis 1** **(H1):**
*There is a non-monotonic relationship between social distancing and self-rated health, where individuals with low and high levels of social distancing experience relatively better health compared to those with moderate levels of social distancing.*


**Hypothesis 2a** **(H2a):**
*According to the sensitizing hypothesis, the negative linear association between social distancing and self-rated health is stronger for individuals experiencing greater food insecurity.*


**Hypothesis 2b** **(H2b):**
*According to the sensitizing hypothesis, the negative linear association between social distancing and self-rated health is stronger for individuals living in communities with limited social capital.*


**Hypothesis 3a** **(H3a):**
*According to the steeling hypothesis, the negative linear association between social distancing and self-rated health is weaker for individuals experiencing greater food insecurity.*


**Hypothesis 3b** **(H3b):**
*According to the steeling hypothesis, the negative linear association between social distancing and self-rated health is weaker for individuals living in communities with limited social capital.*


## 2. Method

### 2.1. Data and Sample

Data were drawn from the three waves (W1-W3) of the COVID-19 Household Impact Survey (“COVID Impact Survey”) fielded in (20–26) April, (4–10) May, and (1–8) June of 2020, respectively, across the U.S. during initial phases of the pandemic [42]. While the survey was conducted at three different time points, the same individuals were not followed over time. That is, COVID Impact Survey does not contain panel data but consists of pooled cross-sectional data. The survey was conducted by NORC (National Opinion Research Center). COVID Impact Survey provides estimates of the American adult household population for 18 regions including 10 States. The sampling frame is based on an extract of the U.S. Postal Service delivery-sequence file, covering approximately 97% of the population. A special user agreement with NORC provided the corresponding author of this study access to the restricted data with 4-digit FIPS (Federal Information Processing Standard Publication) geocodes available at the county level. For the analysis, we pooled the data across W1 (N = 7467; completion rate = 92.7%)), W2 (N = 7420; completion rate = 93.3%), and W3 (N = 6082; completion rate = 92.5%). After listwise deletion of cases with missing values (under 8% of the original sample), data consist of 19,234 respondents. We then selected a representative subgroup of those ages 55 years and over, resulting in the effective sample size of 9124 older adults nested in 645 counties across 18 larger regional clusters (i.e., cities and states). Technical information on sampling procedures and other methodological matters can be found at the online repository maintained by the Data Foundation, a not-for-profit think tank headquartered in Washington D.C. [42].

### 2.2. Measures

#### 2.2.1. Self-Rated Health

General self-rated health (SRH), perhaps the most frequently used indicators of morbidity and mortality [43,44,45], was measured with a single-item question: “*Would you say your health in general is excellent, very good, good, fair, or poor?”* Response options were 1 = Excellent, 2 = Very good, 3 = Good, 4 = Fair, 5 = Poor. Original answers were reverse coded so that a higher number indicates better subjective health.

#### 2.2.2. Social Distancing

To operationalize our main predictor, we used a question asking the study participants about their social distancing and related protective behaviors during the COVID-19 pandemic: *Which of the following measures, if any, are you taking in response to the coronavirus?* 1. Canceled a doctor appointment; 2. Worn a face mask; 3. Visited a doctor or hospital; 4. Canceled or postponed work activities; 5. Canceled or postponed school activities; 6. Canceled or postponed dentist or other appointment; 7. Canceled outside housekeepers or caregivers; 8. Avoided some or all restaurants; 9. Worked from home; 10. Canceled or postponed pleasure, social, or recreational activities; 11. Avoided public or crowded places 12. Avoided contact with high-risk people; 13. Washed or sanitized hands 14. Kept six feet distance from those outside my household;15. Stayed home because I felt unwell; 16. Wiped packages upon entering my home. Original answers (coded 1 if ‘yes’; 0 otherwise) were summed to create an index with relatively good internal consistency (Cronbach’s alpha = 0.71). To test for nonlinearity, we created a squared term for the social distancing measure.

#### 2.2.3. Food Insecurity and Community Social Capital

Food insecurity was measured based on the following survey item: “Please indicate whether the following statements were often true, sometimes true, or never true for you or your household over the past 30 days. [GRID ITEMS]: A. *We worried our food would run out before we got money to buy more*.” Because of the skewed data distribution, original answers were dichotomized (coded 1 for “often true” and “sometimes true”). To operationalize community social capital, we rely on data from “The Geography of Social Capital in America,” a multi-year project funded by the U.S. Congress that ranks counties using the Social Capital Index [46]. The Index consists of 4 sub-indices and 10 variables based on various sources of data collected between 2006–2016. The sub-indices include family-unity (% births to unmarried women; % women currently married; % children with single parent), community-health (non-religious non-profit organizations per 1000; religious congregation per 1000; informal social engagement subindex), and institutional-health (Presidential election voting rate 2012 & 2016; mail-back census response rate; confidence in institutions subindex), and collective efficacy (violent crimes per 10,000). Scores for the sub-indices were first standardized to put them on a common scale and then weights were created for each by running principal components analysis. Each county’s social capital index score was computed by taking the weighted sum of the scores and then standardizing it.

#### 2.2.4. Covariates

Our models adjust for confounding by adding a host of controls at the individual unit of analysis (socioeconomic, demographic, and other variables) that other studies have shown to be related to our focal measures [4,9]. They include age, gender (female = 1) race (non-Hispanic white = 1), education (*1 = “No high school diploma”; 2 = “high school graduate or equivalent”; 3 = “Some college, but no degree”; 4 = “associate degree”; 5 = “bachelor’s degree”; 6 = “master’s degree”; 7 = “professional or doctorate degree”*), household income (*1 = “less than $10,000”; 2 = “$10,000–$ 19,999”; 3 = “$20,000–$29,999”; 4 = “$30,000–$39,999”; 5 = “$40,000 –$49,999”; 6 = “$50,000–$74,999”; 7 = “$75,000–$99,999”; 8 = “$100,000–$149,999”; 9 = “$150,000 or more”*), and family size. Additionally, adjusted are job insecurity (*Think about 30 days from now, how likely do you think it is that you will be employed at that time? 1 = “extremely likely”; 2 = “very likely”; 3 = “moderately likely”; 4 = “not too likely”; 5 = “not likely at all”*), employment status (working = 1), psychological distress (*In the past 7 days, how often have you…[GRID ITEMS]: A. Felt nervous, anxious, or on edge B. Felt depressed C. Felt lonely D. Felt hopeless about the future. 1 = “not at all or less than 1 day”; 2 = “1–2 days”; 3 = “3–4 days”; 4 = “5–7 days.”* Answers were summed to create an index with Cronbach’s alpha = 0.73), comorbidities (*Has a doctor or other health care provider ever told you that you have any of the following? A. Diabetes B. High blood pressure or hypertension C. Heart disease, heart attack or stroke D. Asthma E. Chronic lung disease and COPD F. Bronchitis and emphysema. G. Allergies H. A mental health condition I. Cystic fibrosis. J. Liver disease or end stage liver disease K. Cancer. L. A compromised immune system M. Overweight or obesity Coded 1 if ‘yes’; 0 otherwise*), social connectedness (Answers to the survey items below were standardized and combined as a composite index: *A. During a typical month prior to 1 March 2020, when COVID-19 began spreading in the United States, how often did you talk with any of your neighbors? B. During a typical month prior to 1 March 2020, when COVID-19 began spreading in the United States, how often did you communicate with friends and family by phone, text, email, app, or using the Internet? 1 = “basically every day”; 2 = “a few times a week”; 3 = “a few times a month”; 4 = “once a month”; 5 = “not at all.” C. During a typical month prior to 1 March 2020, when COVID-19 began spreading in the United States, did you spend any time volunteering for any organization or association, or not?* (1 if ‘yes’; 0 otherwise), and, lastly, wave dummies (W1 and W2). At the regional or county level, we control for population density, coronavirus infection cases [47], percentage of Trump-Clinton vote gap during the 2016 Presidential Election [48], the government’s issuing of ‘stay-at-home’ orders [49], latitude/longitude coordinates of the geographic centers (centroids) [50], and the Social Vulnerability Index (SVI) created by the Centers for Disease Control and Prevention which assesses each U.S county as well as Census Tract in terms of area vulnerability based on measures of poverty, unemployment, income, education, population demographics (e.g., proportion of ethnic minorities), vehicle access, crowding, etc. [51]. Table 1 summarizes the details on descriptive statistics.

### 2.3. Analytic Strategy

The COVID Impact Survey data are hierarchically structured, with individual residents nested across regional clusters (counties, cities, and states). That is, the primary sampling units were counties and individual respondents were chosen randomly within those geographic units. As such, respondents living in the same region may be more similar with one another than with those who live in a different region. More specifically, for example, co-residents of a particular community are more likely to be exposed to similar environmental stressors (e.g., coronavirus infection) and hold similar views of social distancing. As such, within regions, there is the methodological problem of correlated errors among individual respondents. This means conventional, i.e., ordinary least squares, regression analysis will produce biased parameter estimates, because its basic assumption regarding the independence of observations is violated. This complex design of COVID Impact Survey thus requires the use of multilevel analysis. Since the data we analyzed provide geocoded information, it allows us to address the issue of data clustering. More importantly, it also enables us to estimate cross-level interactions between the main predictor operationalized at the individual level (social distancing behavior) and the county-level social capital variable for hypothesis testing. Multilevel analysis was conducted using HLM version 8 [52]. Formally, the two-level mixed effects models we fitted take the following forms: (1)$$Y_{ij}=\beta_{0j}+\beta_{1j}Social\;distancing_{ij}+\sum^{\text{​}}\delta_{q}X_{qij}+\varepsilon_{ij},$$
(2)$$\begin{array}{c} \beta_{0j}=\gamma_{00}+\sum^{\text{​}}\gamma_{0s}W_{sj}+u_{0j}\\ \beta_{1j}=\gamma_{10}+\sum^{\text{​}}\gamma_{1s}W_{sj}+u_{1j} \end{array}$$
$Y_{ij}$ in Equation (1) is the predicted value of the outcome (self-rated health); $\beta_{0j}$ is the random intercept varying across regions (counties); $\beta_{1j}$ is the random slope of the main independent variable, *Social distancing*, as defined above; and $X_{qij}$ is the value of covariate q associated with respondent *i* in regional cluster *j*. The error term $\varepsilon_{ij}$ is the individual-level random effect assumed to be independently and normally distributed with constant variance σ^2^. Equation (2) shows our county-level models where $W_{sj}$ is a vector of contextual covariates; and $u_{0j\;}$ and $u_{1j\;}$ are the regional random effects. The covariance between two regional-level random effects is assumed to be zero net of covariates. In the parameter estimation, we apply case weights in the COVID Impact Survey designed for simultaneously analyzing all three waves of data. Table 2 consists of models estimating the relationships between the main predictor and the individual-level moderator (food insecurity). Model 1 examines the hypothesized curvilinear relationship by including individual-level covariates only. Model 2 adds the county-level controls. Model 3 additionally adjusts for comorbidities. Model 4 introduces the level-1 interaction term. Additionally, Model 5 is a robustness check with the inclusion of comorbidities. Table 3 contains models related to the cross-level interaction effect between social distancing and community social capital. Model 1 shows the initial result for the interaction term. Model 2 checks its robustness by adding comorbidities. Finally, Model 3 includes the level-1 interaction between social distancing and food insecurity as a further sensitivity analysis. 

## 3. Results

### 3.1. Testing the Main Hypothesis

Our data consist of 9124 older American adults (ages 55 and over) living across 645 US counties during the initial phase of the COVID-19 pandemic. The counties are in turn nested across 18 regional areas including 10 states (California, Colorado, Florida, Louisiana, Minnesota, Missouri, Montana, New York, Oregon, Texas) and 8 Metropolitan Statistical Areas (Atlanta, Baltimore, Birmingham, Chicago, Cleveland, Columbus, Phoenix, Pittsburgh). With respect to geographical distribution, about ten percent of the sample lived in the Northeast, 27% lived in the Midwest, about 30% were found in the Southwest, and the rest were sampled in the West. Among the respondents, a little over half (54%) were female, about a third (33%) were employed, roughly half (52%) were married, and the majority (84%) were non-Hispanic whites. Forty two percent were between the ages of 55 and 64; forty one percent were between the ages of 65 and 74. The rest (17%) were ages 75 and over. The average income bracket ranged between $40,000 and $49,000. In terms of education, 12% had a high school diploma and 31% had some college education, while 55% possessed a BA or above. 

According to the unconditional model without any of the covariates, there is significant variation in the outcome across contextual units (counties), justifying the use of multilevel modeling to address the issue of data clustering (τ = 0.17; *df* = 733; χ^2^ = 2022.86; *p* < 0.001). The intraclass correlation (ICC) indicates that about 16% of the variance in self-rated health occurs between counties. To test H1 that there is a non-monotonic relationship between social distancing and self-rated health, we initially examine the bivariate relationship without the background controls (model not shown). The parameter estimates for social distancing (*ß* = −0.108, SE = 0.029 *p* < 0.001) and its square term (*ß* = 0.007, SE = 0.002 *p* = 0.001) are both statistically significant. Importantly, the sign shifts from minus to plus: a negative relationship initially exists between social distancing and self-rated health up to a certain point (threshold), after which it reverses the direction. In other words, there is evidence of a U-shape curve. To confirm that the curvilinear relationship is not spurious due to confounding, we add a set of control variables (excluding comorbidities) in Model 1 in Table 2, many of which are predictive of the outcome except employment status, gender (female), race (non-Hispanic white), age and the time (wave) dummies. 

Importantly, while adjusting for potential confounders, we find largely consistent output with respect to the main predictors of interest: social distancing (*ß* = −0.11, SE = 0.029 *p* < 0.001) and its squared term (*ß* = 0.006, SE = 0.002 *p* = 0.002). Model 1, however, does not control for county-level covariates. For a conservative test of our main hypothesis (H1), we include them in Model 2, which shows that, notably, the coefficient for community social capital, SCI, is positive and marginally significant (*ß* = 0.074, SE = 0.04 *p* = 0.065). In other words, net of other county-level measures (population density, coronavirus infection, Trump-Clinton vote gap, state COVID-19 mandate, county-centroid latitudes and longitudes, and structural vulnerability) as well as compositional effects, i.e., individual-level variables, residency in an area endowed with ‘more’ social capital is associated with relatively ‘better’ health. Again, for our main purposes, introducing contextual variables does not significantly alter the effect size or the significance level of social distancing (*ß* = −0.119, SE = 0.029 *p* < 0.001) and its squared term (*ß* = 0.006, SE = 0.002 *p* < 0.001). 

### 3.2. Sensitivity Analysis Using Comorbidities

As a final sensitivity analysis, we incorporate into the model a measure for comorbidities, which would be positively related to social distancing (i.e., preexisting medical conditions would increase the odds of staying at home) and negatively related to the outcome (subjective health). The main finding in Model 3 is that even after adjusting for this critical confounder, in further support of H1, the non-linear association still holds, albeit the parameter estimates for social distancing (*ß* = −0.075, SE = 0.029 *p* = 0.029) and its squared term (*ß* = 0.005, SE = 0.002 *p* = 0.027) are reduced in terms of the effect size and level of significance. Figure 1a (based on Model 4 in Table 2) and Figure 1b (based on Model 5 in Table 2) graphically illustrate the curvilinear association between compliance with social distancing behavior and the health outcome by, respectively, excluding and including the confounding effect of comorbidities.

### 3.3. Interaction Effect with Food Insecurity

To adjudicate the opposing expectations (as stated in H2a and H3a) with respect to the interaction between social distancing and food insecurity, we continue with our findings in Model 4 initially without including the control for comorbidities. The key interaction term is significantly positive (*ß* = 0.056, SE = 0.022 *p* = 0.011). That is, consistent with the steeling hypothesis (H3a), the strength of the negative association between social distancing and self-rated health is weaker among people experiencing food insecurity. As a robustness check, in Model 5, the comorbidities variable is reintroduced. Its inclusion causes the interaction term to fall below the conventional level of significance (*ß* = 0.046, SE = 0.024 *p* = 0.053). Based on this finding, we conclude that H3a receives only partial support. Next, how does community social capital moderate the focal relationship under investigation? To address this issue, we now turn our attention to the models in Table 3 that contain cross-level interaction results. 

### 3.4. Interaction Effect with Community Social Capital

According to Model 1, the coefficient for the interaction term between social distancing and community social capital (SCI) is significantly negative (*ß* = −0.014, SE = 0.007 *p* = 0.033), indicating that in counties with more (less) aggregate social capital the health-damaging impact of social distancing becomes magnified (diminished). In other words, this specific finding is, once again, consistent with the steeling hypothesis (H3b). As was the case earlier, we did not initially control for comorbidities in Model 1. We do so in Model 2 as a sensitivity analysis. The result is consistent: the cross-level estimate is similarly shown to be significantly negative (*ß* = −0.014, SE = 0.006 *p* = 0.024), further confirming the validity of H3b. Figure 2a (based on Model 1 excluding comorbidities) and Figure 2b (based on Model 2 conditioning on comorbidities) are drawn to visualize the moderating role of community social capital. In Model 3, we include the two interaction terms simultaneously as our final robustness check. As shown, the interaction between social distancing and food insecurity (*ß* = 0.042, SE = 0.023 *p* = 0.065) and that between social distancing and SCI (*ß* = −0.013, SE = 0.006 *p* = 0.041) complement our earlier reporting.

## 4. Discussion 

To prevent the spread of COVID-19, social (physical) distancing has been almost unanimously endorsed and adopted as a basic strategy by governments throughout the world. While efficacious in reducing the rate of viral transmission, it has nevertheless produced detrimental side effects: state of social isolation accompanied by loneliness, anxiety, and depression. Not surprisingly, a substantial body of scholarship has emerged documenting mentally harmful consequences of social distancing measures [4,6,7]. Our study sheds novel light on the nature of the association between social distancing and self-assed physical health. Prior research has exclusively emphasized a linear relationship across myriad empirical contexts [4,7]. A rationale behind the linearity assumption is that “physical distancing may increase lack of social connection and, therefore, have a negative impact on physical and mental health” [11]. Based on a probability sample of older American adults, we diverge from this conventional view by demonstrating that the focal relationship is, indeed, non-monotonic or curvilinear. 

As illustrated above, our empirical findings indicate that individuals with a lukewarm (i.e., ‘mid-level’) engagement with social distancing show worse self-rated health compared to those with low or high levels of social distancing, even after adjusting for the confounding effect of comorbidities. Why might this be the case? On one hand, unlike individuals with low levels of social distancing, those with moderate levels of engagement may not have sufficient social integration or support to enhance their health. On the other hand, unlike those who more strictly adhere to anti-pandemic government mandates and thus be protected against the disease, people with moderate levels of social distancing may be fearful or anxious of the viral infection, which in turn may take a toll on their subjective assessment of health. Taken together, our research adds a new perspective to the extant scholarship by documenting a complex (i.e., U-shape) relationship between social distancing and self-rated health. 

Second, our research offers an unexpected, if not ironic, theoretical insight. We juxtaposed two competing mechanisms regarding the moderating roles of food insecurity measured at the respondent level (personal vulnerability) and social capital gauged at the contextual or county level (collective resource). Results for the two interaction terms from our multilevel analysis substantiated the steeling effect thesis, according to which “gathering experience with stressors may enable people to better cope with later stressors” [34]. Substantively, we found that the harmful impact of social distancing is less pronounced for the victims of food shortage, although this interaction became only marginally significant when adjusting for comorbidities. Our interpretation in support of the steeling hypothesis (H3a and H3b) is that instead of undermining individual coping skills, such exposure can sharpen and improve them. This specific finding complements the earlier work, which reveals that early life racial discrimination may foster a steeling effect whereby it helps individuals to develop resilience against other stressors [38]. We suggest that food insecurity may have played a similar role for older American adults. More research is needed to further investigate this seemingly puzzling outcome across different societal and cultural boundaries. 

The steeling effect was also found with respect to the cross-level interaction: the negative association between social distancing and subjective health became diminished in U.S. counties with lower stocks of social capital. In high-social capital communities, as originally recorded in the survey data we analyzed, people by definition interacted more frequently with one another prior to the pandemic. Because of this fact, it is precisely those high-social capital communities that were hit the hardest by COVID-19 and the resulting physical distancing and stay-at-home orders. That is, they produced social isolation and loneliness in ways that were much worse for the (formerly better connected) residents of counties endowed with more social capital. This happened since the pandemic, among other things, undermined the very interpersonal ties and communications that had once served as the basis of social support during times of difficulty. To the contrary, low-social capital implies that community members were more socially isolated and detached from one another to begin with, i.e., before the pandemic. As such, when it emerged, the deleterious consequences of reduced social contact and increased loneliness were relatively less severe. After all, if one already had had limited interpersonal contacts and connections, the negative impact of COVID-19 in terms of reduced social interactions would not have been as dire. In the end, rather than acting as a buffer, community social capital ironically worsened the health-damaging effects of social distancing. 

### Study Limitations and Future Directions

Our research has some limitations. First, the analysis was based on a cross-sectional design, making it difficult to infer causality. While it is plausible for subjective health to shape social distancing behavior, we assumed the opposite flow of influence as reflected in the underlying curvilinear relationship. Given data limitations, however, distinguishing between correlation and causation is not possible in the current context. Hence, we caution readers from making strong causal inference between social distancing and self-rated health, as discussed in our study. Longitudinal or panel data would no doubt help overcome the thorny issue of reverse causation. Additionally, we encourage future researchers to utilize such data, if at all possible, to offer more concrete causal evidence. Second, our study is limited to older U.S. adults. Yet, social distancing may have different meanings across different cultural contexts [53]. It thus remains an empirical question whether our main finding concerning nonlinearity is universal or culture specific. Clearly, this remains an interesting issue worthy of future exploration. Lastly, we could not directly evaluate the specific mechanisms by which food insecurity and living in communities with limited social capital foster a steeling effect. Theory and research suggest that whether steeling effects occur is a function of psychosocial resources (e.g., personal mastery and social support) that individuals draw upon to cope with negative life events [34]. Therefore, future research may benefit from examining the role of psychosocial resources, and, more broadly, assessing when, how, and why steeling or sensitizing effects may operate. 

## 5. Conclusions

In conclusion, at the time of this writing, COVID-19 continues to adversely affect people across the world. In varying degree, but certainly to lesser extent than before, many countries are continually struggling with how best to contain the virus. This paper hypothesized and illustrated that the relationship between social distancing behavior and health outcome among older Americans is both complex (nonlinear) and contingent (conditional on individual and contextual moderators). Despite data limitations, our findings nevertheless strongly hint at such outcomes. Additional investigations would enhance our understanding of these relationships by, first, examining additional measures of “vulnerability” and “resource” as a source of contingency. Additionally, secondly, efforts to unpack the conditions under which the issue of nonlinearity holds valid or not would provide important insights into the double-edged nature of health-protective behavioral prescriptions (e.g., social distancing mandate).

## Figures and Tables

**Figure 1 ijerph-19-16058-f001:**
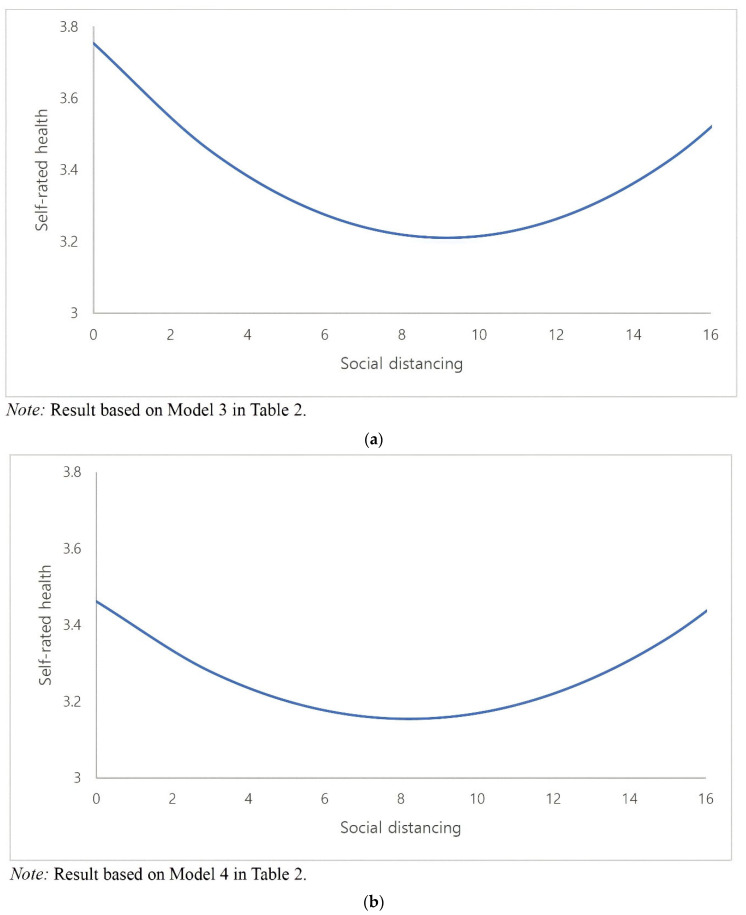
Non-Monotonic Relationship between Social Distancing and Self-Rated Health: (**a**) Not Controlling for Comorbidities (**b**) Controlling for Comorbidities.

**Figure 2 ijerph-19-16058-f002:**
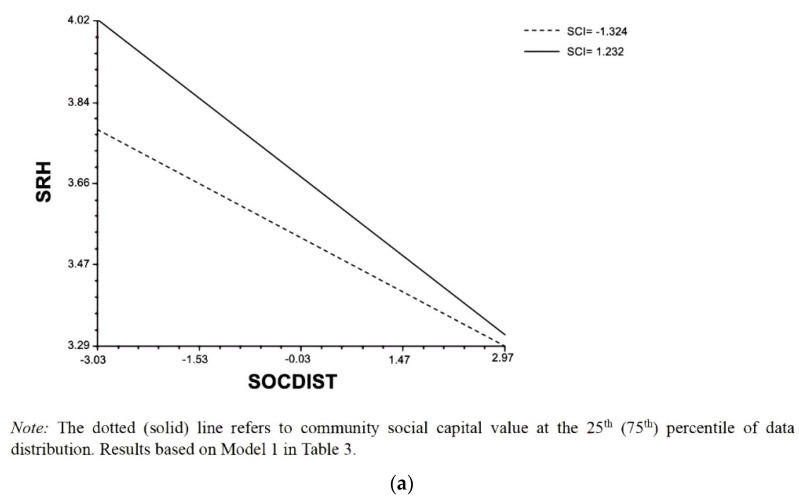
Relationship between Social Distancing and Self-Rated Health Moderated by Community Social Capital: (**a**) Not Controlling for Comorbidities (**b**) Controlling for Comorbidities.

**Table 1 ijerph-19-16058-t001:** Descriptive Statistics, Definitions, and Coding Procedures.

	Mean/Proportion	S.D.	Min.	Max.
Outcome measures				
SRH (self-rated health)	3.63	0.97	1	5
Individual level covariates (N = 9124)				
Social distancing	7.06	2.55	0	16
Food insecurity	11%		0	1
Age 55 (ages between 55 and 64)	42%		0	1
Age 65 (ages between 65 and 74)	41%		0	1
Female (Ref.: male) White (Non-Hispanic)	54% 84%		0 0	1 1
Education	4.45	1.58	1	7
Household Income	5.87	2.27	1	9
Family size	1.88	0.94	1	6
Job insecurity	3.46	1.69	1	5
Employed (1 = currently working)	34%		0	1
Psychological distress	5.71	2.46	4	16
Comorbidities	2.24	1.75	0	13
Social connectednes	0.24	1.87	−7.08	3.70
County level covariates (N = 645)				
SCI (Social Capital Index)	−0.19	0.96	−3.73	2.15
Population density (z-scored)	0	1	−0.17	22.54
COVID-19 cases (z-scored)	0	1	−0.15	25.23
Trump-Clinton vote gap	0.21	0.32	−0.77	0.81
State mandate	14%		0	1
Latitude (county centroid)	37.44	5.68	25.43	48.78
Longitude (county centroid)	−94.32	13.41	−124.2	−68.35
*SVI* (Social Vulnerability Index)	0.51	0.27	0	1

Data source: All individual-level variables are operationalized based on data drawn from COVID Impact Survey (2020). Notes: More details on *SCI* can be found at the homepage of the “Geography of Social Capital in America” project (https://www.jec.senate.gov/public/index.cfm/republicans/2018/4/the-geography-of-social-capital-in-america (accessed on 12 February 2021)) maintained by the U.S. Congress. COVID-19 infection cases are daily tracked by the Center for Systems Science and Engineering at Johns Hopkins University (https://github.com/CSSEGISandData/COVID-19 (accessed on 3 March 2021). Figures for the Trump-Clinton vote gap are available at the MIT Election Lab (https://electionlab.mit.edu/ (accessed on 4 March 2021)). State mandate measures the government’ declaration of ‘stay-at-home’ orders at the time of the survey. *SVI* (Social Vulnerability Index) is constructed by the Centers for Disease Control and Prevention. Technical information on this index is available at https://www.atsdr.cdc.gov/placeandhealth/svi/index.html (accessed on 2 May 2020).

**Table 2 ijerph-19-16058-t002:** Multilevel Analysis Examining the Non-Linear Relationship between Social Distancing Compliance and Self-Rated Health.

DV = *SRH*	Model 1		Model 2		Model 3		Model 4		Model 5	
	Coef.	(SE)	Coef.	(SE)	Coef.	(SE)	Coef.	(SE)	Coef.	(SE)
Intercept	3.620 ***	(0.021)	3.603 ***	(0.025)	3.619 ***	(0.022)	3.602 ***	(0.025)	3.618 ***	(0.022)
(Individual level)										
Social distancing	−0.110 ***	(0.029)	−0.119 ***	(0.029)	−0.075 *	(0.029)	−0.122 ***	(0.028)	−0.078 **	(0.029)
Social distancing squared	0.006 **	(0.002)	0.006 ***	(0.002)	0.005 *	(0.002)	0.006 **	(0.002)	0.004 *	(0.002)
Food insecurity	−0.130 ^†^	(0.067)	−0.154 *	(0.067)	−0.121 ^†^	(0.069)	−0.546 **	(0.169)	−0.440 *	(0.179)
Age55	−0.016	(0.058)	−0.015	(0.059)	−0.051	(0.057)	−0.014	(0.059)	−0.051	(0.057)
Age66	0.021	(0.058)	0.030	(0.060)	0.026	(0.058)	0.030	(0.059)	0.026	(0.057)
Female	0.039	(0.040)	0.038	(0.040)	0.061	(0.038)	0.043	(0.040)	0.064 ^†^	(0.039)
Non-Hispanic White	0.038	(0.056)	0.060	(0.058)	0.057	(0.055)	0.062	(0.059)	0.059	(0.055)
Education	0.071 ***	(0.013)	0.067 ***	(0.013)	0.063 ***	(0.011)	0.069 ***	(0.013)	0.065 ***	(0.012)
Household income	0.088 ***	(0.011)	0.087 ***	(0.011)	0.074 ***	(0.010)	0.087 ***	(0.011)	0.074 ***	(0.010)
Family size	−0.045 *	(0.018)	−0.042 *	(0.019)	−0.047 *	(0.020)	−0.040 *	(0.019)	−0.045 *	(0.019)
Job insecurity	−0.083 ***	(0.016)	−0.081 ***	(0.016)	−0.053 ***	(0.015)	−0.082 ***	(0.017)	−0.053 ***	(0.015)
Employed	−0.005	(0.057)	−0.013	(0.057)	−0.005	(0.048)	−0.011	(0.057)	−0.004	(0.049)
Psychological distress	−0.059 ***	(0.009)	−0.057 ***	(0.009)	−0.036 ***	(0.007)	−0.057 ***	(0.009)	−0.036 ***	(0.007)
Comorbidities					−0.212 ***	(0.011)			−0.211 ***	(0.011)
Social connectedness	0.060 ***	(0.011)	0.058 ***	(0.012)	0.060 ***	(0.011)	0.059 ***	(0.012)	0.060 ***	(0.011)
Wave 1	0.038	(0.036)	0.042	(0.037)	0.038	(0.033)	0.038	(0.037)	0.034	(0.033)
Wave 2	0.018	(0.044)	0.020	(0.044)	0.008	(0.042)	0.018	(0.045)	0.006	(0.042)
(County level)										
SCI			0.074 ^†^	(0.040)	0.050	(0.035)	0.073 ^†^	(0.040)	0.049	(0.034)
Population density			−0.009	(0.019)	0.000	(0.015)	−0.007	(0.019)	0.002	(0.016)
COVID-19 cases			0.013	(0.018)	−0.000	(0.014)	0.012	(0.017)	−0.001	(0.014)
Trump-Clinton vote gap			−0.197 **	(0.074)	−0.122 ^†^	(0.064)	−0.199 **	(0.074)	−0.124 ^†^	(0.063)
State mandate			0.042	(0.062)	0.073	(0.054)	0.046	(0.062)	0.076	(0.054)
Latitude			−0.005	(0.005)	−0.003	(0.005)	−0.005	(0.005)	−0.003	(0.005)
Longitude			−0.002	(0.002)	−0.001	(0.001)	−0.002	(0.002)	−0.001	(0.001)
SVI			−0.004	(0.126)	0.044	(0.112)	0.001	(0.126)	0.047	(0.111)
(Level−1 interaction)										
Social distancing × Food insecurity							0.056 *	(0.022)	0.046 ^†^	(0.024)
Random effects										
Individual level variance	0.631		0.617		0.521		0.614		0.520	
County level variance	0.147 ***		0.145 ***		0.113 ***		0.147 ***		0.113 ***	
Deviance (−2logL)	25,697.559		25,438.091		23,643.178		25,398.734		23,612.560	

Notes: Robust standard errors (SE) are in parentheses. DV = dependent variable. ^†^ *p* < 0.1; * *p* < 0.05; ** *p* < 0.01; *** *p* < 0.001.

**Table 3 ijerph-19-16058-t003:** Results from Multilevel Analysis Estimating the Cross-Level Interaction Effect.

DV = SRH	MODEL 1 A		MODEL 2 A		MODEL 3 A	
	Coef.	(SE)	Coef.	(SE)	Coef.	(SE)
Intercept	3.588 ***	(0.024)	3.606 ***	(0.021)	3.605 ***	(0.021)
(Individual level)						
Social distancing	−0.100 ***	(0.024)	−0.063 **	(0.025)	−0.065 **	(0.025)
Social distancing squared	0.004 **	(0.002)	0.003 ^†^	(0.002)	0.003	(0.002)
Comorbidities			−0.213 ***	(0.010)	−0.212 ***	(0.010)
Controls included	Yes		Yes		Yes	
(County level)						
SCI	0.054	(0.040)	0.027	(0.035)	0.027	(0.035)
Population density	0.008	(0.017)	0.015	(0.014)	0.017	(0.014)
COVID-19 cases	−0.000	(0.015)	−0.010	(0.012)	−0.011	(0.012)
Trump-Clinton vote gap	−0.199 **	(0.072)	−0.114 ^†^	(0.063)	−0.115 ^†^	(0.063)
State mandate	0.049	(0.060)	0.081	(0.053)	0.082	(0.054)
Latitude	−0.004	(0.005)	−0.002	(0.005)	−0.002	(0.005)
Longitude	−0.002	(0.001)	−0.002	(0.001)	−0.002	(0.001)
SVI	−0.043	(0.121)	−0.015	(0.109)	−0.013	(0.109)
(Level-1 interaction)						
Social distancing x Food insecurity					0.042 ^†^	(0.023)
(Cross-level interaction)						
Social distancing × SCI	−0.014 *	(0.007)	−0.014 *	(0.006)	−0.013 *	(0.006)
Random effects						
Social distancing slopes variance	0.011 ***		0.010 ***		0.010 ***	
Individual level variance	0.574		0.482		0.481	
County level variance	0.129 ***		0.103 ***		0.102 ***	
Deviance (−2logL)	25,045.649		23,195.350		23,172.416	

Notes: Robust standard errors (SE) are in parentheses. DV = dependent variable. ^A^ Model adjusts for the individual-level controls as shown in Table 2. ^†^ *p* < 0.1; * *p* < 0.05; ** *p* < 0.01; *** *p* < 0.001.

## Data Availability

The dataset used and/or analyzed during the current study is available from the corresponding author on reasonable request.

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
