# Peer review of "Examining the Complex (Curvilinear and Contingent) Associations between Social Distancing Compliance and Subjective Health during a Global Health Crisis"

_ijerph, 2022, doi:10.3390/ijerph192316058_

Round 1

Reviewer 1 Report

The first paragraph in the Results Section should be devoted to giving a "human face" to the study. The study needs to acknowledge that it deals with human beings, not just numbers. Therefore, this reviewer suggests that the author (s) devote a paragraph or two describing the demographic characteristics of the study sample: age range of the study sample, average age, gender, occupation, education level, income levels, etc, of the study participants.  

Author Response

REVIEWER 1

The first paragraph in the Results Section should be devoted to giving a "human face" to the study. The study needs to acknowledge that it deals with human beings, not just numbers. Therefore, this reviewer suggests that the author (s) devote a paragraph or two describing the demographic characteristics of the study sample: age range of the study sample, average age, gender, occupation, education level, income levels, etc, of the study participants.  

OUR RESPONSE: As recommended, in the Results Section we provided the requested information on the demographic background of survey respondents (see Lines 293-306 in the revised manuscript).

Reviewer 2 Report

The manuscript addresses an interesting topic in public health, i.e., nonlinear associations between social isolation and self-rated health status in the United States during the pandemics.

However, the paper requires revision of English, since some parts are really difficult to understand, particularly the Method section.

First, it would be interesting to point that the three waves of the survey comprise cross-sectional data. The description of the sample design and selection are quite confusing, it could be improved by using straightforward language.

Second, the index for social distancing starts describing the measures within the survey; however, it presents half of the questions and afterwards includes the measures in table 1. Authors should either describe the 16 measures in the text or in the supplementary materials, instead of randomly picking half of the measures for description in the text and then including all in the table.

Third, psychological distress is vaguely described as "a multi-item index" without further details, which is very strange. Again, authors should either include in the text or in the supplementary materials.

Fourth, table 1 includes detailed explanation on most variables (including the phrasing of the questions and the items), which makes the table really hard to read and very confusing. It is better to explain variables in the text (including equations used to estimate each indicator) and include only the variable name in the table.

Fifth, it would be interesting to include one additional column in table 1, indicating the source of each variable.

Sixth, the explanations on the analytic strategy are quite generic and difficult to associate with results obtained in tables 2 and 3. There are five models in table 2 and three models in table 3, but there are no explanations on the differences betweeen diverse models estimated in the study.

Moreover, tables 2 and 3 were included in the Method section, instead of Results, which is very odd.

Results section starts describing non-existing data (page 9, line 247), it would be better to provide the results in the supplementary materials, instead of indicating "not shown".

The Discussion could benefit from additional literature on COVID-19 pandemics and social distancing effects on food security (e.g., https://doi.org/10.1017/S1368980020003493 https://doi.org/10.1002/aepp.13069 https://doi.org/10.1080/19320248.2020.1830221 https://doi.org/10.1007/s40615-020-00892-7 http://dx.doi.org/10.1016/j.clnu.2021.03.018) and social capital (https://doi.org/10.1371/journal.pone.0258021 https://doi.org/10.1177/0890117120924 https://doi.org/10.1007/s40888-021-00255-3 http://dx.doi.org/10.2139/ssrn.3592180 https://doi.org/10.1016/j.socscimed.2020.113365), particularly exploring the results on covariates like race/ethnicity, sex, among others.

Author Response

REVIEWER 2

The manuscript addresses an interesting topic in public health, i.e., nonlinear associations between social isolation and self-rated health status in the United States during the pandemics.

However, the paper requires revision of English, since some parts are really difficult to understand, particularly the Method section.

OUR RESPONSE: We carefully edited the paper throughout to improve the readability, especially the Methods section as suggested.

First, it would be interesting to point that the three waves of the survey comprise cross-sectional data. The description of the sample design and selection are quite confusing, it could be improved by using straightforward language.

OUR RESPONSE: The text was edited to indicate that COVID Impact Survey consists of pooled cross-section data (Lines 145-148). Also, more details were given using clearer language concerning the sample design, data structure, and analytic modeling in the revised text (Lines 255-265).

Second, the index for social distancing starts describing the measures within the survey; however, it presents half of the questions and afterwards includes the measures in table 1. Authors should either describe the 16 measures in the text or in the supplementary materials, instead of randomly picking half of the measures for description in the text and then including all in the table.

OUR RESPONSE: Details are provided in the main text concerning the survey items used for the social distancing variable (please see Lines 173-184).

Third, psychological distress is vaguely described as "a multi-item index" without further details, which is very strange. Again, authors should either include in the text or in the supplementary materials.

OUR RESPONSE: Exact items used to construct this variable, along with others, is stated in the revised text (under 2.2.4 Covariates, Lines 206-232).

Fourth, table 1 includes detailed explanation on most variables (including the phrasing of the questions and the items), which makes the table really hard to read and very confusing. It is better to explain variables in the text (including equations used to estimate each indicator) and include only the variable name in the table.

OUR RESPONSE: Table 1 was edited according to the reviewer’s suggestion by including only the variable names and the descriptive statistics.

Fifth, it would be interesting to include one additional column in table 1, indicating the source of each variable.

OUR RESPONSE: The requested information is stated at the bottom of Table 1 (under Data source)

Sixth, the explanations on the analytic strategy are quite generic and difficult to associate with results obtained in tables 2 and 3. There are five models in table 2 and three models in table 3, but there are no explanations on the differences between diverse models estimated in the study. Moreover, tables 2 and 3 were included in the Method section, instead of Results, which is very odd.

OUR RESPONSE: More information is given in the analytic strategy section (Lines 255-265), as recommended. Tables 2 and 3 were relocated to the Results section. Also, in Lines 281-290, we provided details on each model specification.

Results section starts describing non-existing data (page 9, line 247), it would be better to provide the results in the supplementary materials, instead of indicating "not shown".

OUR RESPONSE: The expression ‘not shown’ was deleted to avoid confusion. All the information necessary from the unconditional model is provided in the main text (Lines 307-310).

The Discussion could benefit from additional literature on COVID-19 pandemics and social distancing effects on food security (e.g., https://doi.org/10.1017/S1368980020003493 https://doi.org/10.1002/aepp.13069 https://doi.org/10.1080/19320248.2020.1830221 https://doi.org/10.1007/s40615-020-00892-7 http://dx.doi.org/10.1016/j.clnu.2021.03.018) and social capital (https://doi.org/10.1371/journal.pone.0258021 https://doi.org/10.1177/0890117120924 https://doi.org/10.1007/s40888-021-00255-3 http://dx.doi.org/10.2139/ssrn.3592180 https://doi.org/10.1016/j.socscimed.2020.113365), particularly exploring the results on covariates like race/ethnicity, sex, among others

OUR RESPONSE: We very much appreciate the above recommendations. We cited the relevant articles in the revised manuscript and included them in the References.

Round 2

Reviewer 2 Report

The suggestions presented in my previous report were included in the manuscript, therefore, I think it is suitable for publication.